# Keep the Right in Mind—A Focused Approach to Right Ventricle-Predominant Cardiogenic Shock

**DOI:** 10.3390/life13020379

**Published:** 2023-01-30

**Authors:** Viana Jacquline Copeland, Alexander Fardman, Ariel Furer

**Affiliations:** 1Leviev Heart Center, Sheba Medical Center, Ramat-Gan, Israel Affiliated to the Sackler School of Medicine, Tel Aviv University, Tel Aviv 6997801, Israel; 2Department of Military Medicine, Faculty of Medicine, The Hebrew University of Jerusalem, Jerusalem 91031, Israel

**Keywords:** cardiogenic shock, right ventricle, right heart failure, hemodynamics, mechanical cardiac support

## Abstract

Cardiogenic shock (CS) remains a highly lethal condition despite many efforts and new interventions. Patients presenting with a rapid onset of hemodynamic instability and subsequent collapse require prompt and appropriate multimodality treatment. Multiple etiologies can lead to heart failure and subsequent shock. As the case prevalence of heart failure increases worldwide, it is of great importance to explore all manners and protocols of presentation and treatment present. With research primarily focusing on CS due to cardiac left-sided pathology, few assessments of right-sided pathology and the subsequent clinical state and treatment have been conducted. This review aims to present an in-depth assessment of the currently available literature, assessing the pathophysiology, presentation and management of CS patients due to right heart failure.

## 1. Introduction

Cardiogenic shock (CS) describes a life-threatening circulatory failure resulting in high rates of mortality, despite a profound advancement in treatments that has been seen in recent years. In fact, with in-hospital mortality reaching up to 50% in some studies [1] and the incidence of CS constantly on the rise, this remains an important clinical condition to understand and study [2].

A wide array of etiologies could lead to the manifestation of CS, including both mechanical and structural abnormalities starting with acute myocardial infarction (AMI), as well as other causes leading to acute and chronic heart failure (HF). While there is an abundance of scientific literature assessing the association between AMI and CS relating to its pathogenesis, management and prognosis, there is relatively scarce knowledge when it concerns other etiologies. In particular, causes manifesting as right heart failure (RHF) resulting in CS have gained less focus, with fewer in-depth assessments conducted.

HF is described as a state of either inadequate cardiac contraction or the impairment of ventricular filling that leads to elevated intracardiac pressure and/or failure of the heart to adequately perfuse organ systems. Some relate to HF and CS as a continuum of the same disease spectrum, even at times approaching CS as an exacerbation of HF. HF could be dissected into subcategories by acuity and by the predominant chamber of the heart that is involved, with each subtype characterized by different symptomatology and prognostic qualities. In fact, this condition is now considered one of the leading causes of morbidity and mortality in the world, with an estimated prevalence of up to 2% in developed countries; this is expected to be on the rise with an aging population globally [3].

Though CS associated with RHF represents only 5% of CS patients, based on recent data, RHF rates are rising. This is thought to be due to the increased life expectancy, the improvement in diagnostic modalities and the increased awareness. Additionally, a substantial quantity of RHF cases are a result of left ventricular (LV) dysfunction, thus further increasing the prevalence [4,5]. It is difficult to estimate the exact prevalence of RHF, as RV-predominant HF remains the final common result in many disease states, and the true prevalence of right ventricular dysfunction (RVD) is therefore most likely underreported [6]. It is thought that, in some HF subtypes, RVD is present in up to 50% of patients, while assessment with cardiac magnetic resonance (CMR) imaging yielded a figure reaching around 19% [7,8]. Though the LV dysfunction associated with CS is more common, representing the majority of cases, right ventricle (RV) damage and subsequent CS are characterized by a more rapid deterioration, reaching hemodynamic collapse earlier than that of LV dysfunction and a higher short-term mortality [9]. However, the RV has better short-term recovery potential, making it critical that patients receive prompt and appropriate treatment, ensuring the opportunity for recovery [8].

In this report, we sought to summarize the contemporary understanding in the scientific literature regarding the role of predominantly right-sided cardiac disease in CS with respect to clinical manifestation, diagnosis, treatment and prognosis.

## 2. Definition and Classification of Cardiogenic Shock

CS is an acute hemodynamic state of hypo-perfusion caused by decreased CO, which results in multi-end-organ damage. CS has been defined in various ways, and one of the most common and practiced definitions arises from the SHOCK (Should We Emergently Revascularize Occluded Coronaries for Cardiogenic Shock) trial, which proposes the following criteria [1]:(1)Hypotension—with a systolic blood pressure <90 mm Hg for at least 30 min or the requirement of supportive measures to maintain the systolic blood pressure at 90 mm Hg.*AND*(2)Cardiac index ≤ 2.2 L/min/m^2^.*AND*(3)Pulmonary capillary wedge pressure greater than or equal to 15 mm Hg.

The SCAI shock classification [2] defines the severity of CS and is based on biochemical laboratory findings, clinical examination and hemodynamic parameters. Patients are classified into one of five stages, ranging between A and E, with stage E being the most severe [10].

The further delineation of CS could be by the dominant side of the heart that is involved, or it could also be described as biventricular shock when the simultaneous failure of both the right and the left chambers occurs [11]. Additional categorization, which is important clinically and guides management, is based on a phenotype with a two-on-two design according to peripheral perfusion (‘hot’ vs. ‘cold’) and pulmonary congestion (‘wet’ vs. ‘dry’) [12].

The diagnosis of RVF is based on the extent of ventricular impairment, using the central venous pressures (CVP), pulmonary capillary wedge pressure (PCWP) [13], pulmonary-artery-pulse-pressure-to-right-atrial-pressure ratios (PAPi) **(**$\frac{PulmonaryArterySystolicPressure-PulmonaryArteryDiastolicPressure}{RightAtrialPressure}$**)** [14] and right ventricular stroke work index (RVSWI) (0.0136×StrokeVolumeIndex×(MeanPulmonaryArteryPressure−MeanRightAtrialPressure)) [15] as indicators [16] (Figure 1).

An additional criteria standard is the Interagency Registry for Mechanically Assisted Circulatory Support (INTERMACS) criteria. RHF, defined by INTERMACS, is based on two parameters: the documented elevated central venous pressure (CVP) and manifestations of elevated CVP in patients with symptoms or findings of persistent RVF. Elevated CVP can be documented using direct measurement with a CVP or right atrial pressure >16 mm Hg, a significantly dilated inferior vena cava with the absence of inspiratory variation by echocardiography or elevated jugular venous distension at least half-way up the neck in an upright patient. Elevated CVP can manifest as peripheral edema (>2+, either new or unresolved), ascites or palpable hepatomegaly on physical examination or by diagnostic imaging, or it can manifest as evidence of worsening hepatic (elevated total bilirubin levels) or renal dysfunction [17].

The Academic Research Consortium definition for RHF further divides patients based on the time of onset and duration. These criteria depict the urgency of the implantation of mechanical assist devices [18].

When classifying CS due to RVF, an alternative form of classification is implemented, based on the underlying physiological mechanisms that cause CS. These include contractility failure, pressure overload secondary to LV dysfunction or other causes of pulmonary HTN, volume overload secondary to right-sided valvular insufficiency and volume overload secondary to an increase in the venous return or following LVAD implantation [16].

## 3. The Right Ventricle—Hemodynamics and Anatomical Characteristics

The high sensitivity of the RV to changes in pressure originates from the nature of the RV, with its high-compliance, low-resistance pulmonary circulation, which makes it suited to adapt to changes in volume rather than pressure [19]. In fact, while RV output approximates that of the left, it is actually attained with a myocardial energy demand of approximately one-fifth that of the left [20]. Pulmonary vascular resistance is estimated to be less than a tenth of the systemic vascular resistance, thus explaining the RV’s milder work force and also its fragility when abrupt changes in pressure occur. It is also thought that the hydrodynamic cycle of the RV differs from the left, lacking an isovolumic period. A phase of isovolumic contraction, where ejection from the RV begins during the pressure upstroke, is hard to define. The RV’s pressure–volume loop lacks the isovolumic phases of contraction and relaxation during systole and diastole [19,21,22,23]. There is a substantial difference between ventricles in the pressure–volume (PV) relationship, as can be seen by a schematic curve of the normal RV. A phenomenon, known as the ‘hangout period’, is unique to the right chamber and is defined by the time lagging from the point in the cycle where the pressure in the pulmonary artery rises above that in the RV and up to the point where the pulmonic valve actually closes. This does not happen with the aortic valve, which closes earlier subsequently to the much higher systemic resistance in comparison with the pulmonary vascular resistance. The meaning of this in terms of the pressure–volume relationship is the practical absence of a clearly defined isovolumic relaxation time. The RV PV curve has a more triangular shape as compared to the square shape of the LV curve [20,22]. Nonetheless, the relationship of instantaneous pressure to volume is as linear as that found in the LV in varying physiological ranges [22] (Figure 2).

Additionally, the ventricular shapes of the two ventricles differ—the LV has an ellipsoidal shape, while the RV has a more triangular shape when assessed from the anterior frontal view and a crescent shape when viewed in a cross-sectional cut. Other factors influence the reduced strength generated by the RV, such as the different consistency of the ventricle compared to that of the left. The RV consists of circumferential and longitudinal orientations, while the LV obliquely arranges myofibers superficially and longitudinal myofibers in the sub-endocardium, with predominantly circular myofibers in between. This arrangement makes the LV more durable, with reduced points of frailty due to the interchanging fiber arrangement [23]. Another aspect would be the thickness of the RV, which is known to be substantially thinner compared to the LV. While the LV wall thickness is between 6 and 10 mm, the RV is 3–5 mm thick (without trabeculations) [25], making it substantially more difficult to generate a counterforce to the pressure and thus weaker. Wall tension is another factor of force that differs among the ventricles, with the thickness influencing the tension capacity. The RV, being thinner, has a milder surface tension, not being able to inhibit backflow during the systole. This would explain the RV’s inability to withstand a high afterload facing a high pressure, but it is very adapted to withstand a high preload and its volume, particularly as it pumps blood to a low-pressure system [23,26]. This would also explain the difficulty in accurately assessing the RV volume, particularly among different stages of the cardiac cycle, yielding overestimating values many times. Common methodologies for calculating the volume include Simpson’s rule and gated blood pool radionuclide angiography [21,27].

It Is important to note that, with the two ventricles sharing a common septum, this union affects the function of each ventricle, particularly that of the right. With the contraction of the oblique muscle of the LV, with a circular rotation and an LV contractile twist, a longitudinal movement of the heart occurs, with an anchoring source of the septum. This sends force to the RV, meaning that the LV is responsible for a portion of the contractile force of the RV.

## 4. Etiology and Pathophysiology of CS

As mentioned above, hypoperfusion is the landmark of cardiogenic shock, and once it is sensed by carotid baroreceptors and juxtaglomerular cells in the kidney, a reflexive sympathetic and neurohormonal response occurs. An increase in catecholamines leads to vascular endothelial constriction. With low perfusion, the renin–angiotensin–aldosterone axis is activated, consequently promoting salt and water retention. These steps result in an increased myocardial afterload and circulating plasma volume. If treatment is not promptly applied, a cycle of decreasing cardiac output (CO) and progressive volume overload ensues, leading to a reduction in coronary artery perfusion pressure, myocardial ischemia, worsening cardiac function and circulatory collapse [23].

When examining an RV insult and its progression to shock, acute damage and subsequent failure lead to a reduction in the LV preload, reduced CO and shock [19]. This is despite normal or only mildly reduced LV contractility. A reduction in RV contractility can also cause reduced right heart output. This is often a result of myocardial fibrosis due to either an infarction or myocarditis. With decreased RV SV, RV dilation occurs, which increases counter-pressure on the tricuspid valve, resulting in tricuspid regurgitation (TR), exacerbates RV dilation and drives a ventricular-interdependent effect on LV filling. This interdependence is described as forces directly transmitted from one ventricle to the other through the myocardium and pericardium [19]. RV dilation and the shift of the interventricular septum toward the LV can damage LV function and exacerbate CS. This is also known as the ‘paradoxical’ movement of the interventricular septum and is accompanied by increased intrapericardial pressure, all of which lead to increased LV end-diastolic pressure and reduced LV transmural filling pressure, depriving LV diastolic filling, which further reduces CO. Subsequently, this state is greatly influenced by the loading of both ventricles (circulating volume), the interventricular septal function and the contribution of the right atrium to RV filling. The latter is very sensitive to disturbances in the normal cardiac rhythm, which can be altered by arrhythmias such as atrial fibrillation. Overall, all of this results in decreased coronary perfusion. Consequently, this state of reduced CO results in tissue hypoxemia and end organ damage [19,27,28,29].

Volume overload can also contribute to RV failure and subsequent CS in special circumstances involving the use of mechanical support devices. A common instance would be a congested state after LVAD implantation. LVAD unloads the LV and derives an increased venous return to the right side of the heart, which can exacerbate pre-existing RV failure [30].

With the volume expansion and the subsequent rise in pressure, the pressure–volume loop of the right ventricle shifts, causing suboptimal ejection and cardiac function (Figure 3).

Right heart involvement in CS could also be approached by dividing the pathophysiology according to the higher afterload the ventricle needs to work against, the declined contractility or the reduced preload of the RV. While, physiologically, the RV has an advantage of pumping blood into a low-compliance pulmonary vasculature, in diseased states such as heart failure, pulmonary hypertension or acute pulmonary embolism, a rise in the RV afterload increases RV pressures and volumes and consequently reduces the RV stroke volume [9]. It is noteworthy to remember that RV afterload is related not just to pulmonary vascular compliance but also to pulmonary parenchymal compliance and intrathoracic pressures [6]. Thus, once congestion occurs, the right overwhelming pressure leads to a reduction in myocardial contractility, insufficient blood oxygenation and output and subsequent congestion [6,32]. It is important to note that LV failure is also a cause of RV failure, mostly affecting it by increasing the RV afterload caused due to the higher LV filling pressures, the lower PA compliance and, therefore, the higher PA resistance and impedance [9]. Notably, the difference between RVD and RVF is that RVD is the anatomical and physiological malfunction of the RV, either due to over-relaxation or hyper-constriction, while RVF is the clinical manifestation of the RVD (Figure 4).

Additionally, it is essential to note that acute insults to an already ailing heart, such as chronic heart failure, can lead to CS. With the gradual increase in pressure and the more indolent decline in stability, the RV can become engorged, particularly in cases where the pericardium is intact. This can subsequently lead to the impairment of the LV chamber, impeding LV filling and equalizing biventricular diastolic pressures. RV systolic and biventricular diastolic dysfunction reduces CO, reducing coronary blood flow and worsening peripheral and abdominal volume accumulation. This could potentially lead to biventricular heart failure. This state is a deadly cycle, as, by disturbing the LV, reduced left-sided heart (LH) filling is more likely to cause RV dilation and ventricular interdependence and further hamper the RV output [19].

### 4.1. Acute RHF

RV–AMI, which occurs acutely, has strong hemodynamic implications, with approximately 25% to 50% of RV infractions being hemodynamically significant due to reduced contractility [33]. The occlusion of supplying coronary arteries causes oxygendeprived cardiomyocytes, leading to tissue damage and necrosis with subsequent alterations in tissue elasticity and contractility [2].

Hemodynamic instability depends upon the extent of the ischemia and infarcted myocardium. Furthermore, elevated filling pressures of the right side of the heart also cause coronary sinus congestion, which deprives the myocardium from coronary blood flow and can provoke RV ischemia and worsen RV function [19]. Subsequent CS development is dependent upon the extent of the damage, with isolated right ventricular infarction representing 5.5% of total CS cases, though some research shows up to 7% [10,23]. This is supported by recent data [34] indicating that isolated RVF is rarer, involving 7% of AMI patients, of which 88% had global RVD on echocardiography and 50% initially had a CVP > 15 mmHg. Additionally, it is thought that a substantial number of total AMI patients have some form of RV impairment, regardless of left-sided integrity. Furthermore, ischemic RVD is observed in up to 50% of patients with inferior AMI. Consequently, acute hemodynamic compromise is evident in less than half of these patients [6]. In the case of acute pulmonary embolism (PE), acute RV involvement is evident in 25–60% of patients [6]. PE occurs as a result of an obstruction in the pulmonary arteries. Obstruction is most often a result of a thrombus originating in the distal venous system, which dislodges and embolizes the pulmonary arterial system. With thrombus sizes differing, obstructive shock can occur due to large occlusions. Some retrospective assessments show that CS prevalence due to acute PE can reach as high as 18.4%. Among patients with acute PE, the total mortality is as high as 15% in the subgroup of patients who present with severe hypotension or cardiogenic shock [35]. Around 8% of acute PEs cause sudden cardiac arrest [36].

Additionally, cardiac tamponade, a pericardial emergency with fluid accumulation, is a worrisome form of CS which can impair RV function [8]. The rapid accumulation of either pus, blood, clots or gas within the pericardial space that compresses the heart leads to a reduction in diastolic filling, preload and contraction [8]. Some causes of cardiac tamponade include transmural myocardial infarction, malignancy, idiopathic, bacterial or tuberculous pericarditis and myocarditis with pericardial involvement [37].

Inflammation of the myocardium, also known as myocarditis, can potentially damage the integrity of the muscle, leading to a decline in RV contractility. There are several causes that lead to myocarditis, most prominently viral, microbial and autoimmune. Viruses represent a major pathogenic factor, which include enteroviruses such as Coxsackie B virus, erythroviruses (most prominently, Parvo B19), adenoviruses and herpes viruses [38]. In the last few years, severe acute respiratory syndrome-coronavirus-2 (SARS-CoV-2) has also caused a prominent disease burden rise due to the COVID-19 pandemic and has been a concerning sequela and complication of the infection. It is thought that approximately 20% of hospitalized patients with SARS-CoV-2 have evidence of cardiac injury, as indicated by elevated levels of high-sensitivity troponin [39]. Generally, it is thought that RV involvement may reflect a greater burden of inflammation compared to LV, though it could also be due to pre-existing vulnerability to an acute process or an increased afterload caused by LHF [19]. Furthermore, a subsequent exposure to the myocardial antigen can progress to autoimmune myocarditis, either post-infectious or idiopathic [38]. In cases of autoimmune myocarditis, some studies have shown that RVD was present in 39% of patients with anti-heart autoantibodies; this is compared to 17% of those without anti-heart autoantibodies [19].

Other infectious causes of acute functional impairment include systemic infection and sepsis, which have been shown to increase pulmonary vascular pressure and propose both to HF and CS [19]. This is thought to be due to cytokine-mediated myocardial depression [40].

Acute LV failure can cause congestion and, in severe cases, lead to acute RV decompensation and shock [8]. Similarly to the causes of acute RHF, the most prominent cause of acute LV dysfunction is a consequence of AMI or the exacerbation of chronic LV conditions [41].

Additional causes, though less prevalent, include various cardiac surgeries that can lead to CS. RV failure in these cases is caused by a volume overload through complications of myocardial ischemia and arrhythmias due to surgery [8]. Furthermore, thyrotoxicosis has also been documented as a cause of acute RHF [42]. Lastly, it has been documented that a rare form of Takotsubo cardiomyopathy solely affecting the RV can cause CS. This condition is thought to be due to emotional strain and can cause cardiac insults [43].

### 4.2. Acute on Chronic RHF

For patients suffering from chronic right HF from various causes (valvular, pulmonary HTN, etc.), even a slight insult can lead to CS, especially among chronic RVD with an elevated pulmonary afterload. A second acute hit due to either an ischemic event, volume overload or intra/post-operative complications could definitely lead to RV-predominant CS [19].

## 5. Signs and Symptoms

The diagnosis of CS is a moltimodility-dependant process, combining clinical symptomatology, physical findings and hemodynamical and echocardiographic assessement. Using these tools, an acurate portrayal of a clincal state can be achived (Figure 5).

### 5.1. Patient Presentation

The general presentation of CS patients tends to be unpredictable, though undeniably in severe distress. Patients are most commonly described as ‘cold and wet’, referring to cold extremities and pulmonary congestion. This is usually coupled with oliguria (urine output < 30 mL/h), dizziness and mottled skin. Once severe, confusion and alterations in one’s mental status can be present [29]. Symptoms also depend upon the etiology of CS, as acute and chronic diseases appear differently, both physiologically and clinically.

Nonetheless, in patients with CS due to an RV MI, the presentation varies, with some remaining asymptomatic, while others experience severe hypotension. Patients present with dyspnea less frequently and more commonly have general malaise prior to medical intervention [34]. Notably, in isolated RVF, lung sounds are clear, and oxygen saturation is normal [23].

CS events with a prominent cardiometabolic phenotype (i.e., obesity, diabetes insulin-resistant glucose metabolism, dyslipidemia, and hypertension) are more likely to have right heart dysfunction and liver injury. This is particularly true for patients suffering from more chronic RHF and subsequent CS. Cardiogenic cirrhosis includes the spectrum of hepatic disorders that occur secondary to hepatic congestion due to cardiac dysfunction, most prominently originating in the RV [46]. The symptomatology can present as right upper quadrant discomfort due to hepatic congestion, which is one of the first signs of RV failure [6]. Additionally, vascular gastrointestinal congestion can arise, with systemic congestion halting abdominal function, leding to reduced intestinal absorption and a damaged intestinal barrier [46,47].

### 5.2. Physical Examination

Examinations can yield several findings. Jugular venous distention with prominent v waves, an evident hepatojugular reflux and peripheral edema can be present. In severe cases, anasarca can develop. Additionally, hepatomegaly and splenomegaly can develop due to systemic congestion. When auscultating, a holosystolic murmur of tricuspid regurgitation might be present. Though the blowing murmur may be absent in patients with acute RVF, a hepatic pulse and signs of concomitant LV dysfunction can be heard. Furthermore, a paradoxical pulse can be felt [8,40].

### 5.3. Hemodynamic Alterations

Overall, hemodynamic change and instability are prominent characteristics of CS. Hypotension, a common feature of CS, is present in an overwhelming majority of patients, though other hemodynamic alterations can occur as well, such as a narrow pulse pressure [23]. Though less prevalent, normotensive shock represents about 5% of cases in the SHOCK trial registry, having an SBP of over 90 mmHg due to supranormal systemic vascular resistance and a low cardiac index. Nonetheless, CS patients with right-sided heart failure are more likely to present with hypotension [23]. When assessing the central venous pressure, a rise in pressures is common; this is to ensure a sufficient flow across the pulmonary circulation. Depending upon the circumstance and etiology, an increase in PCWP can occur, evidently due to left-sided pathology, which consequently damages the right and pulmonary embolism. Furthermore, once congestion becomes severe, an increase in right atrial pressure (RAP) can be seen. One of the common subsequent exanthems is jugular venous distension, which also can be seen. Due to decreased right CO, decreased LV filling from impaired ventricular interdependence is seen [48,49,50].

The pulmonary artery index pressure (PAPI) is a crucial parameter used to assess RV function, and it is defined as the ratio between pulmonary artery pulse pressure and right atrial pressure. A PAPI of <1.0 is a highly sensitive indicator of RV failure in the setting of an acute myocardial infarction [16]. RV stroke work is a tool used to estimate RV function, though the calculation of RV stroke work requires a true estimate of CO, which is commonly measured with the Fick method in RV failure. Thermodilution is thought to be inaccurate, as tricuspid insufficiency is a common exanthem in RVD. The simplest approach to quantifying RVD is to measure the ratio of right atrial (RA) to pulmonary capillary wedge pressures [9]. An RAP of more than 15 mmHg, an RAP and PCWP ratio higher than 0.8 and an RV stroke work index (RVSWI) of under 600 mmHg mL/m 2 are all indices of RV failure [51].

Notably, the cause of CS can also lead to different hemodynamic alterations. Heart failure, pulmonary hypertension or acute pulmonary embolism can all result in a rise in RV afterload and increased RV pressures and volumes followed by reduced RV SV [16]. HF patients have a significantly higher mean pulmonary artery pressure and peripheral vascular resistance compared to AMI patients, leading to a significantly lower RA/PCWP ratio, while the PAPI values are significantly higher [10].

### 5.4. ECG

Alterations in electrical activity are expected due to the potential damage to the atrioventricular and sinoatrial nodes. This is more often followed by supraventricular arrhythmias and sinoatrial or complete atrioventricular blocks [52]. Additionally, atrial flutter or fibrillation can also be present [47]. Patients with right infarction can present with ST-segment elevation in lead V4R that is greater than 1 mm, which is stated to be 100% sensitive. Additionally, R:S ratio in lead V5 or V6 ≤ 1, S in lead V5-V6 ≥ 7 mm, P-pulmonale can be seen, with specificity thought to be between 83% and 95%. [39,46] P-pulmonale can be seen, with a specificity thought to be between 83% and 95% [47,52]. Patients with PE might show evidence of the ‘RV strain’, an incomplete right bundle branch block, an ‘S1Q3T3′ pattern and T-waves in V1 through V4, which are associated with an eightfold increased risk of mortality [40]. Furthermore, some alterations can serve as clues regarding the timeline of the disease, with chronic RV failure often having right-axis deviation due to RV hypertrophy [47].

### 5.5. Laboratory Findings

Due to hypoperfusion and systemic congestion, subsequent sharp increases in circulating transaminases and lactic acidosis are particularly more likely to be present among patients with RV damage and CS due to RV impairment [6,11]. With hepatic congestion, aside from the raised transaminases and high bilirubin, a prolonged prothrombin time can also be present [8]. Some less specific parameters indicative of cardiogenic damage include quantities of Cystatin C and interleukin-6. These findings can shed light on the HF extent and heart muscle integrity [53,54]. Additionally, classical cardiac biomarkers such as troponin I and T- or B-type natriuretic peptide (BNP) can also be elevated, depending upon the etiology. Among cases of PE, a heightened D-dimer is noted [8].

Cardiorenal syndrome, a state of acute decompensated heart failure leading to kidney injury, could accompany right-sided CS. An increase in CVP and a consequent rise in renal vein pressure can cause kidney congestion with heightened blood urea nitrogen levels and creatinine [47,55].

## 6. Investigation Modalities

### 6.1. Non-Invasive Modalities

Trans thoracic echocardiography is the most common non-invasive tool used to assess RHF; it is carried out by assessing the motion of the RV and the PA pressure using RV dynamic characteristics. While a RV basal diameter of over 4.2 cm is indicative of RV distension and increased pressure, additional hemodynamic alliteration occurs. An RV fractional area change of under 35%, a tricuspid annular plane systolic excursion of less than 16 mm, a systolic annular velocity of under 10 cm/s and an RV ejection fraction of under 44% are indicative of impaired RV function [6]. Furthermore, the presence of a tricuspid regurgitation, with a TR velocity of over 2.8 m/s, could indicate high RV systolic and PA pressures. Ventricular akinesis is an additional feature representing ventricular failure [44,45].

Due to the curved shape and structure of the RV, volumes are challenging to evaluate in an objective manner [56]. Some discrepancies among echocardiography exist, as it is user-dependent and has differing accuracies depending upon the site assessed [6]. Due to the acute setting of CS, a transesophageal echocardiogram is not routinely used. Furthermore, 3D-echocardiography and strain imaging have proven to be useful and accurate imaging modalities, but they are not as widely used. Other modalities such as cardiac magnetic resonance imaging and cardiac computed tomography (CT) are less helpful in the CS setting or in the bedside management of RV-predominant shock [6], but CT can shed light on RV alterations and hemodynamic and collateral damage. These can include RV hypertrophy, right atrial enlargement or pulmonary artery enlargement and validate the presence and extent of PE, if present [40].

### 6.2. Invasive Modalities

Pulmonary artery catheterization (PAC) or right heart catheterization (RHC) enable comprehensive hemodynamic assessment. RV injury, leading to a reduction in the RV peak systolic pressure, an increase in the RV end-diastolic volume, a reduction in the RV stroke volume (SV) and a subsequent afterload increase, is seen. These pressures can be assessed using PAC [16].

RHC’s benefit has been contested, with the ESCAPE trial not demonstrating a clinical benefit or shorter hospitalization in stable, hospitalized patients, and this is thought to be due to the invasive nature of the procedure. On the contrary, other assessments have shown a reduction in admission mortality and 30-day readmission with the use of RHC-guided therapy in cardiogenic shock patients [57]. This, in turn, emphasizes the need to identify appropriate candidates who can undergo these procedures [10,57,58]. Both the AHA and ESC emphasize that RHC is not routinely recommended for all patients with decompensated HF; yet, in cases of decompensated HF refractory to the initial therapy, involving patients with an undefined volume status and ongoing hypoperfusion or who are in need of advanced therapies, its use is indicated [51]. Therefore, the use of RHC should be strictly kept for complex and severe cases where the RAP/PCWP ratio is associated with adverse clinical outcomes. Early invasive hemodynamic assessment may lead to an earlier and more accurate diagnosis of CS. This is generally true of all patients with signs of hypoperfusion and suspected CS [23]. Though few assessments of RHC among patients with RHF have been conducted, the consensus is that RHC has prognostic competence in the acute diagnosis of RHF. Patients with PHT or RHF resulting from LHF can benefit from RHC, particularly among patients who underwent orthotopic heart transplants [52].

## 7. Management

The approach to the management of right-side-predominant CS needs to take into consideration the etiology of shock, with an attempt to directly treat a reversible cause, when possible. In addition, the management will be guided by hemodynamic evaluation and be in accordance with the presumed pathophysiology. The spectrum of interventions includes the volume administration, vasoactive drugs, diuretics, agents that enhance cardiac contractility and invasive intervention with mechanical support devices in cases of refractory shock (Figure 6).

### 7.1. Pharmacological

**Inotropic agents**: Generally, over 90% of patients with CS receive some form of vasoconstrictive medication [23]. These constrictive vascular agents, which are particularly useful in patients with low blood pressure, act by enabling an increase in blood pressure and perfusion by increasing the peripheral vascular resistance. Additionally, patients with CS and RHF are treated with vasopressors and inotropes to improve coronary perfusion [6].

The first group of drugs is inotropic agents with a vasoconstrictive effect, including agents such as epinephrine and norepinephrine alongside dopamine at high doses. The use of vasopressors in patients with RVD and CS is advocated in order to maintain blood pressure and ensure perfusion. This is particularly true in patients with a systolic BP <80–90 mm Hg [19]. Generally, in patients with CS, norepinephrine has been shown to have a superior efficacy compared to epinephrine [59,60], though a specific assessment of CS due to RVD is not available. Norepinephrine can restore systemic hemodynamics without having any effect on the pulmonary vascular resistance or RV afterload [6]. It has been seen that patients with RV failure with sepsis could benefit from treatment with norepinephrine, leading to RV myocardial oxygenation improvement but at the cost of a rise in peripheral vascular resistance. Works in animal models suggest a direct effect not only on the vascular bed but also on intrinsic cardiac contractility [61]. Additionally, improvements of ventricular systolic interaction and coronary perfusion are seen [59,60]. Dopamine has a higher propensity to cause more arrhythmic events [16]. Overall, for patients with RV failure and persistent hemodynamic instability despite the optimization of RV loading conditions, inotropic therapy with these agents has been shown to improve the total CO [16]. It is important to note that, despite the optimization of RV loading conditions, persistent hemodynamic instability may still endure; subsequently, these patients may need further inotropic therapy with either a phosphodiesterase inhibitor or a β1-adrenergic receptor agonist that may further improve the total CO [16].

This second group of agents includes drugs such as dobutamine, levosimendan and phosphodiesterase III inhibitors (milrinone), agents that have both ionotropic and vasodilatory effects. These agents should be used with great caution, as they can precipitate hypotension, though in cases when LV function is intact, they can simultaneously improve contractility and increase CO and are thus indicated in patients with RV-predominant HF causing CS [6,19,62].

Agents in this group have been shown to have equivocal hemodynamic properties. Levosimendan and milrinone may favorably affect ventricular–arterial coupling by combining RV inotropy and pulmonary vasodilation, and they might be preferentially indicated in patients with pulmonary hypertension caused by left heart disease. For patients with severe pulmonary hypertension, milrinone is usually the preferred agent for reducing pulmonary-artery pressures and improving RV function [62,63,64]. Milrinone is less likely to cause tachycardia and may be a treatment option in a setting of concomitant β-blocker therapy. However, it is more likely to cause hypotension [64]. Prolonged treatment with these agents is not recommended, particularly with milrinone due to the risk of developing tolerance. When considering treatment options, it is important to take the drug’s half-lifetime into consideration. Milrinone has a long half-life, meaning it is active for longer durations of time, and its effect will continue for longer after withdrawal. Furthermore, milrinone excretion is partially renal-dependent, and, thus, the glomerular filtration rate needs to be examined before treatment initiation. This is contrary to dopamine, for example, which has a short half-life and a rapid onset and is easily withdrawn when ceased. Hypotension is less likely with Dobutamine, and it has less of an effect on the RV afterload [62,65].

Though the superiority of different drugs has been assessed, randomized controlled trials failed to show any superior effects of either drug. Additionally, an overwhelming majority of the research focuses on CS due to LV pathology. A more in-depth assessment of the two drug groups among patients with CS due to RVD is needed in order to find the optimal drug protocol and sequence. 

**Vasopressin**: Unlike the previous agents, vasopressin has no inotropic properties and does not improve the cardiac power index and CI [59,60]. Arginine vasopressin, which causes peripheral vasoconstriction, with less of an impact on pulmonary vascular resistance, has beneficial effects in avoiding tachyarrhythmias while supporting renal perfusion [6]. Many consider vasopressin as the second-line treatment for patients with RHF and hypotension [61]. Vasopressin has a beneficial effect on renal function and is indicated in patients with potential renal damage. This is due to selective efferent arteriole constriction, aiding glomerular filtration. Additionally, it improves the mean arterial pressure and coronary artery perfusion and reduces the risk of RV ischemia and further damage. This agent is particularly useful in cases of refractory hypotension caused by peripheral vasodilation, which is not respondent to primary treatment [65]. Caution should be taken when using vasopressin, as high doses may cause pulmonary vasoconstriction through the sensitization to circulating catecholamines [61].

**Volume maintenance:** As RV-predominant shocks are preload-dependent, the optimization of CVP and the management of the total body volume status are essential. The goal of reaching an optimal RV preload is a very challenging task and requires close hemodynamic evaluation. As a result, therapy could include either volume expansion or diuretics, at times aiming to maintain CO without worsening venous congestion [9].

Adding fluids often depends upon the LV filling pressure, with low parameters increasing the risk of prerenal azotemia. Furthermore, in cases of adequate renal function, the urine output can serve as an indicator of whether fluids are needed. If the risk of renal damage due hypovolemia is feared, fluids could be given, though cautiously [65]. However, it is important to remember that salt retention occurs during states of hypotension due to the activation of the renin–angiotensin–aldosterone system, thus also causing fluid retention. Being precise in giving fluids is of great importance, with the fear of congestion and edema due to fluid overload [65]. Furthermore, it has been seen that empirically giving fluids can cause hemodynamic deterioration [61].

As a result, diuretics are frequently used as the first-line agents, reducing volume overload and systemic congestion. The extent of treatment is etiology- and comorbidity-based, with many patients having renal impairment [6,16]. Most commonly, loop diuretics are used, which are very efficient at achieving the desired effect. If patients are unresponsive to the initial treatment, a continuous infusion of loop diuretics is required to maintain the decongestive effect, with an increase in more efficacious dosages [6].

The elevated renal venous pressure often found in such patients contributes to decreased renal blood flow and reduces the perfusion pressure, which decreases the diuretic efficacy. Monitoring can be carried out by assessing the urinary sodium content [6,16]. In cases in which the urine output is inadequate despite adequate diuretic maintenance or in which renal failure occurs, renal replacement therapy should be considered.

**Pulmonary vasodilator therapies:** Pulmonary hypertension is a condition often seen together with HF and can be a cause of RV failure due to primary vascular pathology or secondary to other conditions such as pulmonary embolism or left heart pathology or failure. In any of these cases, RV afterload reduction is important in negating RV distention [9].

Furthermore, it is shown that pulmonary vasodilators may improve RV CO while not increasing the pulmonary arterial pressure and reducing the RV afterload, though more data on this topic are needed [16,59,60].

Epoprostenol, a vasodilator, has long-term clinical benefits, including an improved survival and functional capacity for patients with pulmonary HTN. It is given either in an inhaled form or intravenously, with the latter being more potent. Nitric oxide is another agent that has potent vasodilatory effects and is regularly used in this subset of patients. Both of these agents have been shown to improve oxygenation and pulmonary vascular resistance. Due to the rapid action of the intravenous modalities, they are used in acute and intensive care settings [61].

Furthermore, administering vasodilators to patients with pulmonary hypertension has to be carried out with great caution, as different subtypes can respond differently. Among patients with features of venous or capillary involvement (veno-occlusive disease and pulmonary capillary hemangiomatosis), vasodilators and prostacyclins have been shown to be dangerous and increase mortality [66,67].

Caution should be taken when considering the administration of vasodilatory therapy in the context of CS, as a further decrease in systemic blood pressure may occur, with subsequent potential reduction in the RV preload and worsening RV ischemia. This, in turn, can worsen CS due to an RV insult [6].

A novel treatment is the Omecamtiv agent, a selective cardiac myosin activator targeting the myotrope. Though its primary use has been for patients with HF with reduced ejection fraction (HFrEF), it has shown promising trial results aimed at RVD management. With promising results, 20 weeks after the trail initiation, an improvement in RV regarding the RV systolic ejection time, RV outflow tract velocity time integral, RV end-systole area, RV afterload and RV–pulmonary artery coupling was seen among these patients. This can be a potential treatment option for patients with HFrEF who are experiencing an exacerbation of a hemodynamic state [68]. 

### 7.2. Invasive

**Percutaneous coronary intervention (PCI):** This invasive intervention has a role when a reversible cause such as obstructive coronary disease is deemed to be the cause of RV-predominant CS [9]. Early and successful revascularization results in less extensive RV myocardial necrosis and is associated with better outcomes. Proper revascularization is critical, with a TIMI flow under two in at least one RV branch being independently associated with myocardial necrosis [65].

Multivessel revascularization or culprit vessel intervention is an additional consideration when assessing a treatment. It is thought that culprit vessel intervention has a significant benefit over multivessel revascularization, though research assessing its efficacy among RVD is needed [65]. As in any case of AMI, emergency revascularization of the infarct-related artery remains the mainstay of treatment and is the only therapy that has significantly reduced mortality in CS to date [23]. Among patients with RV MI and subsequent CS, right coronary artery (RCA) occlusion has been found in an overwhelming percentage of patients (97%). with the second being the left circumflex artery (LCX) (3%) [34]. When assessing the precise location of occlusion in the RCA, it has been seen that the majority of events occur proximally to the origin of the RV branches. If the occlusion is in the left anterior descending (LAD), the LAD route most often proceeds around the LV apex, terminating at the inferior wall [65].

**Interatrial shunt**: Among patients with CS with acute RV failure and PAH that are not responding to medical therapy, a bridge to transplantation (typically for the lung) in the form of a surgically created passage in the atrial septum could be employed. A balloon atrial septostomy (BAS) provides a right-to-left shunt to unload the RV. Additionally, though not always implemented, a BAS might be followed by a Potts shunt formation procedure. A distinct approach creates a shunt between the left PA and the descending aorta [19].

This procedure will most certainly derive a decrease in systemic oxygenation and thus must be outweighed by the increased oxygen delivery mediated by the increased CO. The identification of appropriate patients is needed, and the criteria include: patients with severe RHF or an RAP > 20 mmHg, significant hypoxemia (<90% on room air) or a pulmonary vascular resistance index > 4400 dynes·s·cm−5/m^2^. Furthermore, proper preoperative preparation is of great importance. The optimization of filling pressures and periprocedural inotropic support could be needed [19].

With a similar rationale, it was observed that a natural patent foramen ovale has been associated with more favorable hemodynamics and improved survival, though further research is needed [6].

**Thrombolysis and thrombectomy:** Among patients with CS due to PE, thrombolysis or thrombectomy could be indicated. Thrombolysis is intended to break down the thrombus using anticlotting agents, primarily the tissue plasminogen activator (tPA). This is in contrast to thrombectomy or surgical embolectomy, which are forms of mechanical thrombus removal [69]. Systemic thrombolysis is recommended if the systolic blood pressure is <90 mmHg or the bradycardia is <40 beats/minute, according to the American Heart Association. According to the European Society of Cardiology, systemic thrombolysis is advocated in cases of high-risk PE and should ideally be given within the first 48 *h* after the symptom onset, though variations can be considered. If contraindicated, patients can be treated with surgical embolectomy, or percutaneous catheter-directed treatment can be provided [70,71].

Additional considerations include worsening respiratory insufficiency, severe RVD or major myocardial necrosis, all along with low risk of bleeding complications [71]. Surgical pulmonary embolectomy or catheter-directed thrombectomy are indicated in patients with contraindications to fibrinolysis or with persistent hemodynamic compromise or RVD despite fibrinolytic therapy. As of late, new, less invasive modalities, using more advanced modalities, have been used to retrieve the thrombus. The Indigo Thrombectomy System (Penumbra, Inc, Alameda, CA) is a system which uses a continuous vacuum pump to engage and withdraw the thrombus. Additionally, the FlowTreiver aspirates the thrombus using a nitinol disk; this pierces the thrombus and eases extraction. As these modalities are new, further clinical research is needed to assess their efficacy [72,73].

**Pericardiocentesis:** In cases such as extensive pericardial effusion or tamponade, an echo-guided pericardiocentesis might be indicated. This is dependent upon the etiology, the size and the effect on hemodynamic stability. Though the size of the effusion is not always indicative, the presence of tachycardia, tachypnea, pulsus paradoxus and echo-Doppler features including cardiac chamber collapse and the increased respiratory variation of transvalvular flow velocities could indicate a life-threatening state [37].

### 7.3. Supportive Treatment

**Oxygenation:** Proper oxygenation is critical in trying to prevent intubation for as long as possible. Supplemental oxygen is recommended in patients in order to maintain oxygen saturations greater than 92%. This is to avoid hypoxic pulmonary vasoconstriction and increases in the RV afterload. Additionally, metabolic and respiratory acidosis should be monitored and corrected in order to prevent an increase in pulmonary vascular resistance [40]. Proper oxygenation is of particular importance among patients with COPD, as appropriate oxygenation can not only help in stabilizing patients but also improve RV hemodynamics. High flow nasal cannula (HFNC) or nebulized beta agonists can be of great benefit in these patients [65].

**Mechanical ventilation:** For CS patients who are hemodynamically unstable or have difficulty breathing, mechanical ventilation could be indicated, though this has to be employed with caution in patients with RHF. Mechanical ventilation may negatively impact both the RV preload and afterload, worsening shock. This is due to the difference between spontaneous and mechanical respiration physiology. Spontaneous inspiration leads to negative intrathoracic pressure transmitted to the right atrium and enhances the venous return. This is in contrast to positive pressure ventilation and positive end-expiratory pressure (PEEP), which have the opposite effect. Though no specific ventilation strategy among CS patients with RHF is recommended by guidelines, the damaging effects of mechanical ventilation on RV function can be diminished by the precise use of PEEP and by limiting the tidal volume to avoid lung over-distension [6].

### 7.4. Percutaneous Mechanical Circulatory Support Device (PMCS)

In cases where non-invasive modalities are not sufficient, early mechanical circulatory support (MCS) should be considered as a measure for preventing multi-organ or BiV failure [74]. This is achieved by using temporary MCS in acute settings, particularly when surgery seems to not be appropriate at the time, enabling stabilization and providing time to facilitate a definitive procedure or intervention. Depending on the intervention, this time can be described as ‘bridge to support’, ‘bridge to transplant’ or, ideally, ‘bridge to recovery’, meaning cardiac recovery and consequent successful weaning [74].

A spectrum of devices is available, each with a different specialty structure and mechanism. As the heart hemodynamics differ between sides, machines are adjusted to properly fit each side and guarantee optimal hemodynamic aid. They can be divided into direct RV bypass and indirect RV bypass systems. Direct systems, such as Impella RP and TandemHeart RVAD, move blood from the RA to the PA, thus directly bypassing the RV. This is in contrast to systems such as VA-ECMO, which moves and oxygenates the blood from the RA to the femoral artery, thus indirectly bypassing the RV by skipping the PA and the respiratory system and dispersing throughout the body [9]. Several recent reports support the concept of an early MCS support initiation time and have observed improved survival with the early initiation of short-term MCS in the setting of AMI-CS [74].

**Intra-aortic balloon pump (IABP):** This computer-controlled device is mainly used in acute conditions once CO is severely impaired [75]. IABP is less effective in situations of acute RV failure, with its primary action influencing the LV [30]. Nonetheless it may aid by unloading the LV and eventually reducing right-sided filling pressures. Additionally, it can be helpful by increasing right coronary perfusion, though it is important to note that studies have not portrayed strong beneficial effects in this clinical context [76]. This is confirmed when examining animal models with RVF [77]. Generally, among patients with RVF, IABP was shown to improve hemodynamics, even within minutes after the insertion; this was particularly pronounced within the first hour. Though with a prolonged treatment duration, the improvement rate was seen to slow down. Among these patients, IABP use is associated with reduced inotropic drug use and improves peripheral tissue perfusion. This is thought to be due to improved coronary and myocardial perfusion [77]. In cases where IABP is thought to have failed, this is considered to be due to an RV pressure overload [78].

**Impella:** This intracorporeal micro-axial MCS device is primarily used for patients with CS, it being placed across the aortic valve into the LV. Blood is actively pumped from the LV into the aorta, thereby unloading the heart and increasing CO. The active unloading of the LV reduces the myocardial load and oxygen demand, improving coronary perfusion and increasing peripheral tissue perfusion [79].

Several types of Impella are currently offered, including the Impella RP, which is exclusively used for right heart support. The device inflow portion is located in the IVC or close to the junction with the right atrium and the flexible nitinol pump is advanced into the outlet portion at the pulmonary artery, thus significantly increasing RV CO and reducing the central venous pressure following the placement and increasing MAP. It has been seen to have a more beneficial effect in improving hemodynamics compared to IABP [80]. Even once removed, the beneficial effect is maintained, though not to the same extent. The Impella RP can be used for up to 14 days and can provide a flow of up to 5 L/min [16,27]. The RECOVER RIGHT trial demonstrated the safety, feasibility and efficacy of the Impella RP in patients with RV failure due to acute MI or after LVAD implantation, with flow rates ranging between 2 and 5 L/min, significantly improved CI values (from 1.8 ± 0.2 to 3.3 ± 0.23 L/min/m^2^ (*p* < 0.001)) and reduced CVPs (from 19.2 ± 4 to 12.6 ± 1 mm Hg (*p* < 0.001)) [6,81]. Additionally, Impella support in patients with right AMI, post-unsuccessful RCA PCI and subsequent RVF, has been shown to aid in immediate hemodynamic stabilization (an increase in systolic blood pressure from 91 ± 17 to 136 ± 13 mm Hg), decrease the CVP (from 16 ± 2.5 to 12 ± 4 mm Hg), reverse the shock and improve the survival at 30 days [82].

Common serious complications of using the Impella are the risk of bleeding, which is four times higher than that in control groups, and peripheral vascular complications, which are twice more prevalent [79]. Caution should be implemented when using the system inpatients who have TR or PR, and though this has been contested, some reports suggest that functional tricuspid regurgitation caused by the dilation of the valvular annulus may improve with Impella RP treatment. Furthermore, there are exclusions for the Impella RP in the presence of prosthetic valves or tricuspid valvular anomalies [81].

### 7.5. Right Ventricular Support Device (RVAD)

These types of devices are intended for use for a longer period of time and work by blood withdrawal from the right atrium or by ventricle pumping it to the main PA, based on an extracorporeal energy source. Its use is indicated when cardiac function is not predicted to improve to allow for a bridge for further interventions and to promote functional improvement. It has been used in a wide array of cases from severe RV MI to severe pulmonary hypertension, severe mitral regurgitation, allograft failure following heart transplantation as well as post-LVAD implantation [30]. Several systems are offered for such treatment, with different mechanical circuits, all of which have their benefits and downfalls.

**TandemHeart:** The Tandem Heart (TH) (Tandem Life, Pittsburgh, Pennsylvania) is an extracorporeal centrifugal pump and a continuous flow percutaneous mechanical assist device that can be used to treat extreme, refractory CS due to acute RVF. Two venous cannulas are introduced: an inflow for drainage of the right atrium and an outflow cannula that is placed in the pulmonary artery. The cannulation of these cannulas is usually executed through the left femoral and the right femoral veins, respectively [74]. According to the THRIVE registry, improved CO, increased MAP, reduced CVPs and reduced pulmonary pressures with mean flows of 4.2 ± 1.3 L/min with the TandemRVAD were reported [6,16]. The benefits this system offers are the avoidance of sternotomy and the percutaneous approach, as well as eliminating the need for a perfusionist [6].

However, complications were evident in clinical trials and included cannula migration causing acute hemodynamic collapse, RV arrhythmia, perforation as well as persistent bleeding around the cannulation site and in the GI tract and hemolysis. Further, an increased risk of stroke and cardiac tamponade was seen among these patients [9,80].

**ProtekdDuo:** This device is very similar to the previous one but has a unique dual lumen cannula that can enable single venous access via the right internal jugular vein for use with the TandemHeart-RVAD [83]. The use of Protek Duo has most commonly been reported in RV failure following LVAD implantation and in the setting of pulmonary hypertensive crises leading to RV failure [30]. A flow of 4–5 L/min can be obtained, and the jugular cannulation allows for ambulation. Reports on its efficacy are based on very small cohorts, with some studies showing 23% patient weaning success but a more than 40% mortality rate despite the proper pump flow [30].

**CentriMag:** Similar to the previous devices, the CentriMag (Thoratec, Pleasanton, CA, USA) is an extracorporeal RVAD with a centrifugal pump and a motor, but it is implanted surgically. Its unique design is thought to reduce blood trauma and mechanical failure such as clotting and hemolysis [83]. Compared with other devices, this system tends to have a high flow mean, with the flow reaching 10 L/min The inflow cannula is inserted directly to the right atrium through the superior or inferior vena cava, or it could be inserted further into the RV. The outflow cannula is typically anastomosed to the PA. It could be incorporated together with an ECMO device and also include an oxygenator. Evidence was summarized in a meta-analysis of patients treated with the CentriMag due to indications such as postcardiotomy shock, post-transplant allograft rejection and RV failure following LVAD placement, with survival rates at 30 days of 41–66%, depending on the clinical condition leading to RVAD use. It is currently approved for use as an isolated RVAD for up to 30 days in patients with RV-related CS [30]. Obviously, its advantages are countered by the need for surgery and its consequent complications [30].

When compared retrospectively, percutaneous RVAD use was associated with reduced morbidity, including a decreased blood transfusion requirement and a shorter time spent being mechanically ventilated. This would support the use of pRVAD among patients who are expected to need a more intensive treatment to support their RV and CS management. Ventricular assist devices enable patients to stay mobile during treatment and many times reinstitute daily activity. Nonetheless, currently, no FDA-approved long-term implantable devices designed to maintain RV support are available [30,83].

It is not uncommon that RVADs are used as part of biventricular assist devices in cases where both ventricles are incompetent. This is particularly common among patients with RV MCS as a consequence of durable LVAD placement. These patients tend to have very poor outcomes, with less than half of patients remaining stable after the removal of temporary RVAD support. For these patients, the HeartWare (Thoratec, Pleasanton, CA, USA) was available; a ventricular assist device was developed which was mostly used for patients awaiting a heart transplant, though it was discontinued in 2021 due to high thromboembolic events. Additionally, HeartMate 3 (Thoratec, Pleasanton, CA, USA) from Abbott can be adjusted to function as a biventricular assist device, even in the highly trabeculated RV. Nonetheless, treatment places the patients under a significant risk of thrombosis development [30].

**Veno-Arterial Extracorporeal Membrane Oxygenation (VA-ECMO)**: This system is composed of a circuit oxygenating blood, bypassing the heart and the lungs with a venous inflow cannula, a centrifugal flow pump, an oxygenator, a heat exchanger and an outflow arterial cannula with either central or peripheral employment methods [71]. Providing full cardiac support, full respiratory support and protection during life-threatening arrhythmias and being able to completely replace the native heart function with flows of up to 6 L/min, it is a particularly useful tool and is increasingly used for short-term support in CS [74], irrespective of RV or LV function, by establishing a parallel circulation [30]. VA-ECMO is most commonly considered in patients with profound cardiogenic shock and in the setting of cardiac arrest. It is used when the treatment is expected to be prolonged, it being particularly useful when end organ damage is present [9]. In acute RHF secondary to obstructive diseases of the pulmonary vasculature, VA-ECMO, rather than an RVAD, is the preferred choice, as the increased pulmonary blood flow from the RVAD could further increase PAP [19]. VA-ECMO displaces and oxygenates blood from the RA to the femoral artery, bypassing the RV, reduces RV preload and RV CO while increasing systemic mean arterial pressure and LV afterload [9,30,71]. In cases in which there is isolated RV failure, VA-ECMO will reduce RA and PA pressure initially and decrease the LV preload. As a result, the LV afterload will increase substantially, leading to the presence of preserved LV function, with CO which may remain unchanged or decrease. It is important to note that VA-ECMO may not always be the ideal tool in cases of biventricular failure, where VA-ECMO will decrease RA and PA pressure and LV preload, yet due to the poor LV function, the increased LV afterload will reduce CO, increasing LV filling pressures. This subsequently induces pulmonary edema and results in an increase in the mean PA pressure [9]. When assessing the efficacy of VA-ECMO in cases of acute RV failure, very little evidence exists in the literature, with the impact of VA-ECMO on right-sided heart hemodynamics still not fully explored. Among patients with refractory CS, due to either AMI or post-cardiotomy shock, the mean central venous pressure is reduced by 5 mmHg, and the mean PA is reduced by 11 mmHg after 24 h of treatment. Thus, it can be interpreted that VA ECMO could have salutary hemodynamic effects in patients with RV failure [9,74].

Treatment with VA–ECMO comes with its risks, some of them being life-threatening. Some of the most dreadful ones are thrombosis, with estimates reaching a rate of 21.5%, and lower extremity ischemia, with estimates being as high as 24% [84]. The first could result in distal embolization events with a manifestation as PE and stroke, among others, and mandate the administration of anticoagulants, which may lead to further complications [85]. The risk of profound bleeding, which is estimated to be present in about a quarter of patients with some sort of major bleeding complication, can happen in patients, even regardless of anticoagulation therapy [30,85]. Additional adverse events can follow the use of VA-ECMO and include severe inflammatory responses and the worsening of pre-existent pulmonary edema. Scores have been developed to assess the likelihood of these complications, such as PRESERVE (predicting mortality for severe ARDS on VV-ECMO), SAVE (Survival After Veno-Arterial ECMO) and the simple cardiac ECMO scores [85].

An alternative to VA ECMO is the veno-veno ECMO (VV ECMO), which is thought to be the preferred system in cases of isolated RV failure secondary to acute hypoxemic respiratory failure. It takes deoxygenated blood and returns oxygenated blood to the venous system. VV ECMO is known to reduce PA pressures, improve CI and reduce the central venous pressure (CVP). Furthermore, its further benefit is by mediating a decrease in hypoxia-mediated pulmonary vasoconstriction, along with improvements in PaCO2 and pH, resulting in increased cardiac contractility (CC) [83].

**Anticoagulation and MCS devices:** Meticulous anticoagulant therapy is needed throughout the treatment duration to maintain a thrombus free circuit, required in all RVAD systems. The first-line option is heparin, monitored with the activated clotting time (ACT) or partial thromboplastin time (PTT), with a target of about 1.5 to 1.9 times the laboratory normal levels. Second-line agents include bivalrudin or argatroban, with a potential consideration of warfarin treatment. With excessive treatment, bleeding can occur, which is usually quickly corrected with blood products such as platelets and cryoprecipitate. If weaning attempts are carried out, protamine can be used to reverse effects [30].

## 8. Future Directions

Currently there are several new treatment modalities aimed at improving treatment efficacy and patients’ outcomes. The PERKAT^®^ RV pulsatile right ventricular support device augments the efficacy of the IABP device, with a pump chamber connected to an outlet tube and a pulmonary trunk, bypassing the right heart. Though showing a promising efficacy, tests have been limited to animal models, showing an increased cardiac output by 59% in sheep suffering from acute RHF [86,87].

Another focus point of novel treatment is venous decongestion. The Doraya Catheter is a temporary intravenous flow regulator installed in the inferior vena cava. This is thought to improve hemodynamic parameters and renal function outcomes. With the first human study showing promising results and safety approval, further research and advances are bound to occur in this field [88].

An additional venture is the Circulite device (HeartWare Inc.), a fully implantable hybrid axial-centrifugal–flow pump moving blood from the RA to the PA. Testing among patients with RV failure and pulmonary hypertension is currently in progress. With potential positive results, the further improvement in hepatic and renal congestion may evolve [9].

## 9. Prognosis

Generally, CS, being the most extensive form of heart impairment, naturally has a very poor prognosis, with an in-hospital mortality rate of up to 50% [89]. Aside from the well-established exacerbating factors of CS, such as severe hypotension and end organ damage, the RVF-associated CS has additional prognostic features. Though research assessing its complications is lacking, several studies have reported poorer clinical outcomes in patients with RV failure in the setting of left-sided heart failure [16]. Due to the fact that the RV is susceptible to a rapid decline in function, a short timespan is needed for the transition from stable conditions to shock [90]. Furthermore, patients with predominantly RV failure had a significantly lower 30-day mortality compared to patients with LV failure—35% vs. 50% [34]. However, these data are contested by others stating that mortality among RV-associated CS is equivalent to that of LV.

It is clear that the prognosis of patients is dependent on the disease severity. As a result, several score modalities were designed in order to assess patients’ potential for clinical instability and 30-day mortality probability. One of these is the CLIP score, which assesses cystatin C, lactate, interleukin-6 and N-terminal pro B-type natriuretic peptide (NT-proBNP) levels at admission. Though its specific benefit among patients with RV failure CS has not been established, it has shown great efficacy generally among patients with CS [90]. An additional scoring system is the SCAI, which is thought to have a high prognostic value, while being easy to utilize. Based on hemodynamic features including hypotension, tachycardia and general patient stability, patients are classified into one of five stages, ranging from ‘At risk’ or ‘A’, this being the mildest form, to ‘Extremis’ or ‘E’, the most severe form. Additionally, by adding to the SCAI score with invasive hemodynamic measurements and echocardiographic findings, an additive prognostic value is observed. It has been suggested that the three-axis observation system can be used as an assessment modality, further improving the accuracy of the SCAI score. This is due to its broader assessment criteria, spanning the examination of shock severity, the clinical phenotype and risk modifiers of both the RV and LV [91,92,93,94]. Nonetheless, neither’s efficacy has been documented among patients with RVF-related CS.

On the bright side, a profound improvement in RV function in survivors of CS caused mainly by RVD is often seen, particularly following the RV infarct. A better prognosis and clinical progression are seen among patients who are recognized to have RVD early, along with treatment aimed at the quick relief of RV ischemia [90]. Recent advances in percutaneous therapies for short-term RV circulatory support offer a source of improvement among patients with notoriously poor outcomes [30].

## 10. Summary

CS caused by right heart dysfunction is a cause of cardiovascular morbidity and mortality that has gained less scientific focus compared with CS due to LV dysfunction. Various etiologies can cause RVD and the subsequent deterioration to CS, with the treatment being greatly dependent upon the pathological source. With RHF rates increasing and expected to rise in the upcoming years, this topic needs further research assessing the current protocols, treatment management modalities and mechanisms while simultaneously developing new methods of reducing morbidity and mortality that are more phenotype-specific for RV failure.

## Figures and Tables

**Figure 1 life-13-00379-f001:**
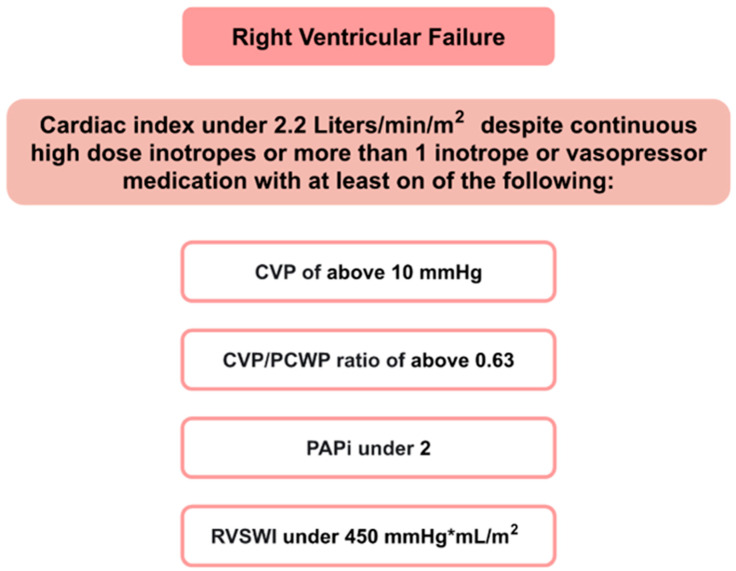
Criteria of right ventricular failure [16].

**Figure 2 life-13-00379-f002:**
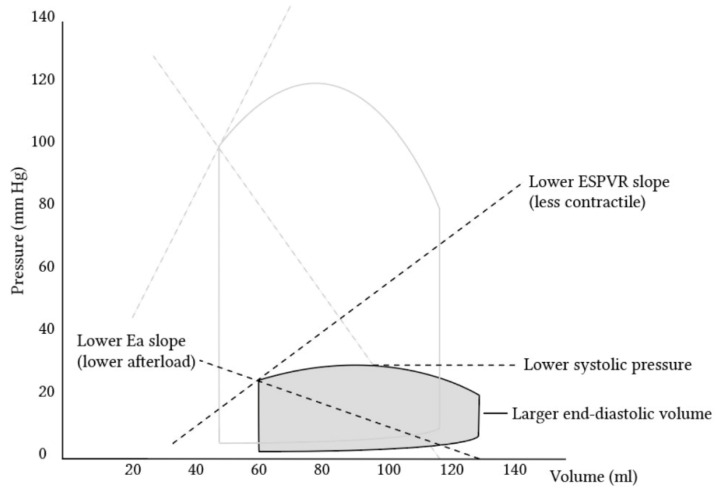
Pressure–volume loops comparing the left ventricle (transparent) and right ventricle (gray). It is visible that the right ventricle creates substantially lower pressures, among other differences. (Adopted with permission from https://derangedphysiology.com by Alex Yartsev [24]).

**Figure 3 life-13-00379-f003:**
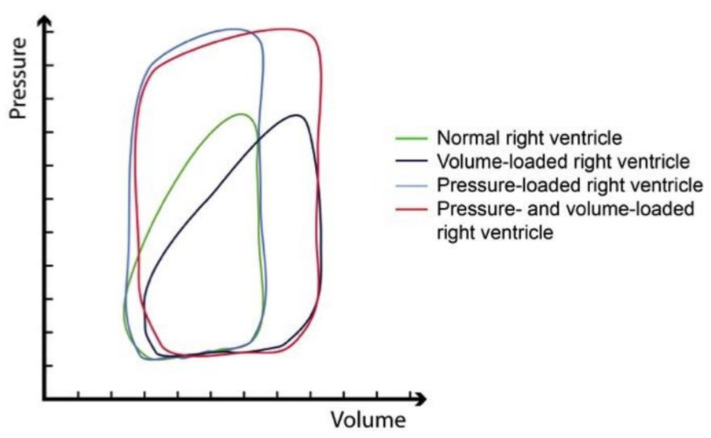
Pressure–volume loops of the right ventricle in a normal state (green); conditions of volume overload (dark blue); condition of pressure overload (light blue); mixed pressure and volume overload (red). Notably, while the shape of the curve changes in a very minor manner with higher volumes, the curve changes substantially and resembles that of a left ventricle when the right ventricle is required to cope with higher pressure. (Adopted with permission from De Meester et al. [31]).

**Figure 4 life-13-00379-f004:**
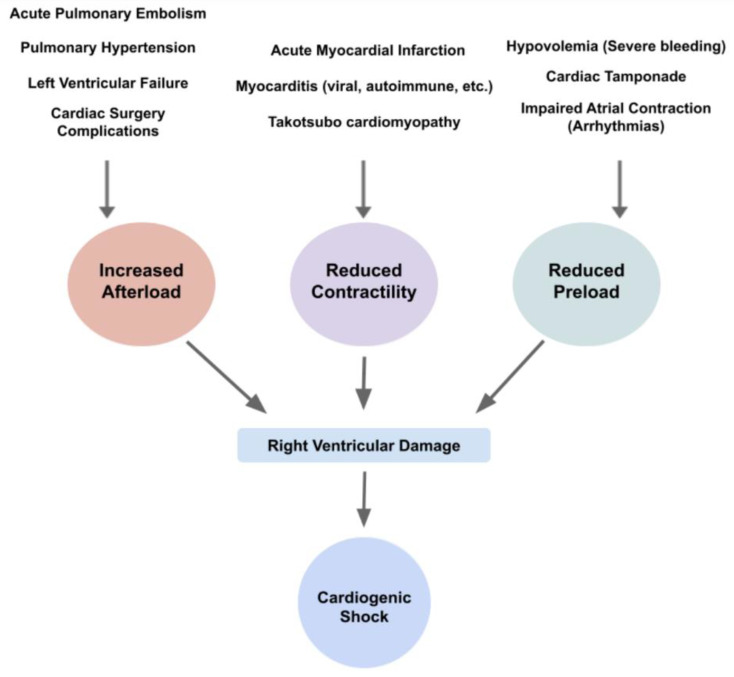
Diagram of etiologies and their physiological changes, leading to right ventricular damage and subsequent cardiogenic shock.

**Figure 5 life-13-00379-f005:**
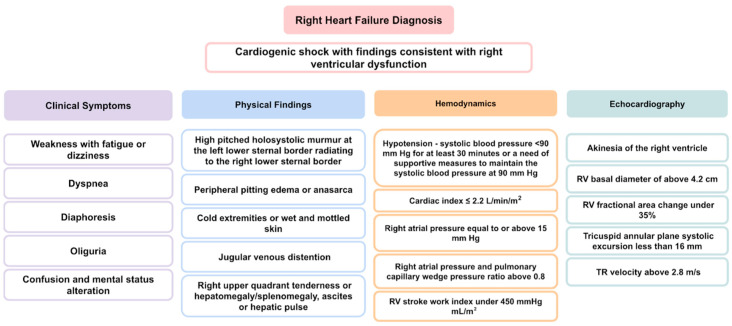
Diagnostic features of cardiogenic shock resulting from right heart failure. The features included consist of clinical symptoms, physical findings, hemodynamic parameters and echocardiography findings [16,44,45].

**Figure 6 life-13-00379-f006:**
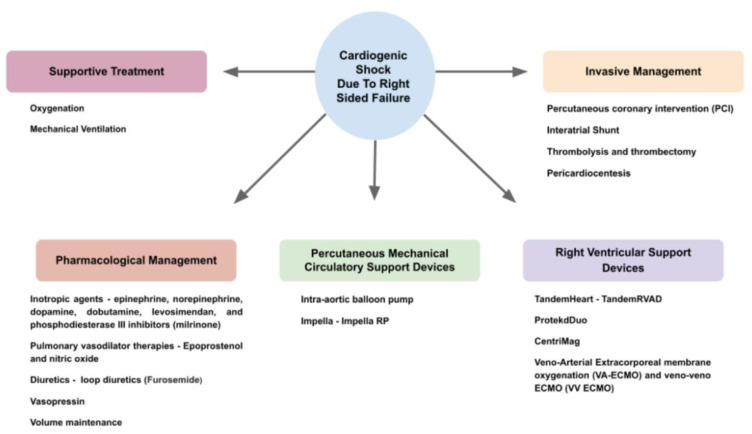
Diagram of the treatment modalities and their entities available for the management of cardiogenic shock due to right-sided failure.

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
