# Peer review of "Keep the Right in Mind—A Focused Approach to Right Ventricle-Predominant Cardiogenic Shock"

_life, 2023, doi:10.3390/life13020379_

Round 1
Reviewer 1 Report
The review is well written, the authors presented a clear description of the mechanisms responsible for right heart failure, therapeutic strategies, especially about invasive procedures and future strategies, all accompanied by an exhaustive review of the recent literature.
Some minor issues:
I would suggest a more detailed definition of right heart failure, for example comparing the INTERMACS definition with the MCS-ARC one, together with the hemodynamic criteria that you've already described;
About hemodynamic criteria (eg. PAPi, RVSWI), consider to insert their mathematical definition (formula)
I would also suggest a more complete native English grammar revision;
Author Response
Attached below is a file with our reply letter.
Please see the attachment.

Reviewer 2 Report
This is a comprehensive review on cardiogenic shock from right heart failure. Authors included detailed literature review for definition, etiology, pathophysiology, manifestations, diagnostics, therapeutics and prognosis of right heart predominant cardiogenic shock. A few questions and comments are as below:
line 45, what is the cause of rising rates of RHF? And a reference for this trend is needed.
line 64. Definition of cardiogenic shock should include end organ hypoperfusion. Hemodynamic criteria should fulfill both cardiac index and PCWP , therefore should include “and” between 1) and 2).
line 85, Figure 1 is visualized but also should be explained in the manuscript with citing Figure 1. This applies to all figures. Only Figure 2 was cited.
lines 375-378 are written in bold characteristics.
line 378. Another subtitle with “Signs and symptoms”. (see line 289)
lines 408-420 include explanation regarding all CS. Is there any specific data regarding the benefit of RHC for right heart failure?
Author Response

(The authors gave the same response as above.)
